# Sorghum Flour Application in Bread: Technological Challenges and Opportunities

**DOI:** 10.3390/foods11162466

**Published:** 2022-08-16

**Authors:** Pervin Ari Akin, Ilkem Demirkesen, Scott R. Bean, Fadi Aramouni, Ismail Hakkı Boyaci

**Affiliations:** 1Field Crops Central Research Institute, Ankara 06170, Turkey; 2Department of Food Engineering, Hacettepe University, Beytepe, Ankara 06800, Turkey; 3Department of Animal Health, Food and Feed Research, General Directorate of Agricultural Research and Policies, Ministry of Agriculture and Forestry, Ankara 06800, Turkey or; 4Center for Grain and Animal Health Research, USDA-ARS, 1515 College Ave., Manhattan, KS 66502, USA

**Keywords:** sorghum, bread, gluten-free, tortilla, flatbread, pan bread

## Abstract

Sorghum has a long history of use in the production of different types of bread. This review paper discusses different types of bread and factors that affect the physicochemical, technological, rheological, sensorial, and nutritional properties of different types of sorghum bread. The main types of bread are unleavened (roti and tortilla), flatbread with a pre-ferment (injera and kisra), gluten-free and sorghum bread with wheat. The quality of sorghum flour, dough, and bread can be improved by the addition of different ingredients and using novel and traditional methods. Furthermore, extrusion, high-pressure treatment, heat treatment, and ozonation, in combination with techniques such as fermentation, have been reported for increasing sorghum functionality.

## 1. Introduction

Sorghum (*Sorghum bicolor* L. Moench) is a grass belonging to the Andropogoneae tribe of the Poaceae family (alt. Gramineae) and is among the top five cereals in the total world production after wheat, maize, rice, and barley [1]. Sorghum is a warm season cereal with a C_4_ photosynthetic pathway. Sorghum is a drought-tolerant crop and has advantages over similar crops such as corn in areas of limited rainfall [2]. Climate change is anticipated to impact the production and composition of cereal crops, especially through drought and heat stress [3,4,5]. The selection of heat and drought-tolerant sorghum lines has been recommended as one avenue to mitigate the effects of climate change on cereal production [6,7,8,9]. Sorghum is also an important crop for risk management for producers. Sorghum yield was found to be less susceptible to environmental influences than corn in the Great Plains region of the US; therefore, it may help mitigate production risks [2]. Sorghum’s resilience to heat and drought stress and its ability to grow in marginal lands or areas where other crops do not produce well are also important for global and regional food security [5,10,11].

Sorghum is similar to other cereal grains in overall proximate composition. However, genetic variability, environmental influences, crop management, and abiotic/biotic stresses can all cause significant variation in grain composition, and variation among sorghum germplasm may be significant due to the rich genetic diversity of germplasm [12]. Accordingly, sorghum grain may contain a wide range of proximate values [13,14,15,16]. Sorghum also has a significant amount of minerals such as phosphorus (P), zinc (Zn), iron (Fe), calcium (Ca), magnesium (Mg), potassium (K), and sodium (Na), as well as vitamins such as B vitamins, vitamin E, β-carotene, etc. [14,16]. In addition to meeting basic nutritional needs, sorghum grain is also known to contain bioactive compounds that have beneficial properties such as hypocholesterolemia, anti-inflammation, anti-cancer, anti-obesity, and anti-diabetic activity [17,18,19]. The most important of these bioactive compounds are phenolics, which include unique 3-deoxyanthocyanins that are not commonly found in high concentrations in other cereal grains [18]. They also have a greater percentage of slowly digestible and resistant starch components as compared to other staple cereal crops. The pericarp color of the sorghum grain causes differentiation in phenolic components in the sorghum grain, which can vary widely from white to various shades of red, yellow, and black depending on both genetic and environmental factors [16]. Some sorghum types, only those with a pigmented testa layer, also contain tannins, which have a strong antioxidant capacity and unique human health attributes [16,18]. On the other hand, like any grain, sorghum also contains anti-nutritional components such as phytic acid, trypsin inhibitors, and other compounds that may have detrimental consequences. Several technological processing methods, including germination, fermentation, cooking, soaking, and steaming, can reduce antinutritional components [19]. Such processing technologies allow sorghum grain and derivatives to be used as food additives in beverages such as beer and in bakery products [1,18]. These processing methods can also increase the quality of sorghum grains and resultant products such as sorghum starch, bread, and cake, etc.

Although sorghum grain has been used as a staple food for significant numbers of people, especially in the sub-arid tropics, it is increasingly used in gluten-free food products and as a food ingredient in other regions of the world due to its newly discovered health-beneficial properties [10]. In areas or markets where sorghum is less expensive than other cereal grains, sorghum may be used in place of corn in various foods, as well as other traditional flours, to reduce food costs.

Among the other foods, bread has been a basic food item throughout human history.

Bread types and production processes vary greatly throughout the world. The term “bread” can be used to describe a wide range of products which have different shapes, sizes, textures, and crusts; however, the essential ingredients of bread are cereal flour, water, yeast or another leavening agent, and salt. Other non-essential ingredients typically included in bread formulations are sugar, enzymes, emulsifiers, oxidants, fat, and additives. Furthermore, bread-making processes can be divided into three basic steps which are mixing, fermentation, and baking. In mixing or dough formation stage, all ingredients must combine well. In the fermentation step, this step allows the dough to rise and develop desired crumb structure. In the baking stage, the dough expands further and the bread takes its final shape [20,21]. For many years, sorghum flour has been used in bread, especially in flat breads such as roti, injera, and kisra. Since sorghum proteins lack the unique functional properties of wheat proteins in terms of viscoelastic dough formation, sorghum flour has been used in flat breads for many years. With the rising global interest in gluten-free grains and functional breads made with various flour combinations, sorghum has started to be incorporated in pan breads. The current literature review specifically focuses on traditional and novel sorghum-based bread products of all types, the challenges faced in developing such products, and the advantages already achieved with the use of sorghum grain.

## 2. Methodology

For the current literature review, the types of available evidence in a given area were identified, the key definitions in the literature were clarified, and the research methods on this topic were examined.

First, literature databases including PubMed, Web of Science, and Science Direct were searched for keywords including sorghum, bread, traditional, gluten-free, tortilla, flat and pan bread, kisra, roti, and injera, which were combined using the Boolean operators (and, or). In terms of traditional sorghum bread products, almost all relevant manuscripts’ findings were included in the present study. However, in terms of sorghum gluten-free bread and wheat bread, mostly studies conducted after the 2000s were included. Papers were selected on an inclusive basis as the literature was limited in terms of sorghum used in bread and bread-like products, especially related to the use of 100% sorghum.

## 3. Flat Breads

### 3.1. Roti

Roti, which is commonly consumed in Western and Central India, is an unleavened bread typically made with pearl millet or sorghum flour [22,23,24]. Sorghum roti has various names in different parts of India: chapati (Hindi), bhakri (Marathi), rotla (Gujarati), rotte (Telugu), etc. [25]. To produce roti, sorghum flour and the optimum amount of water are mixed until they reach the preferred consistency. The kneaded dough is divided into small balls which are flattened with the help of a rough wooden or metal surface, before then being dusted with flour and baked on both sides on a hot plate [24]. A wide range of research has been conducted in relation to various factors that govern roti quality and production (Table 1).

To evaluate the quality of roti, two different aspects were used: (a) dough handling properties, or how easily the dough stretches to make thin pancakes, and (b) sensory attributes such as softness, chewing properties, dryness, and absence of bitterness [44].

Pericarp color, endosperm type, and texture significantly affect the quality of roti. The most preferred grains in roti preparation are white/pale yellow, dense, and round grains [38]. Murty et al. [38] indicated that roti made from sorghum with 100% corneous endosperm had a firm texture and lower storage quality, whereas those with a floury endosperm had unfavorable dough properties with poor flavor and storage quality. The use of corneous grain up to a certain content improved dough and roti quality. With regards to endosperm texture, corneous grains had higher grain density, grain-breaking strength, and less water absorption. Good roti-producing grain varieties had a colorless thin pericarp, 60–70% corneous endosperm, less than 24% grain water absorption, and a flour particle size index of around 65 [38]. The particle size index of sorghum flour varied between different varieties and was linked to the endosperm texture of the grain. Grinding/milling techniques also had significant impacts on sorghum flour characteristics, including particle size and starch damage (Table 1).

Grain structure (degree of corneous endosperm, kernel weight, breaking strength and water absorption), dough properties (water required, kneading quality, rolling spreading), and roti sensory characteristics (color, taste, texture, aroma, storage) were all affected by the environment, season, year, and genotype and year interactions [38]. Roti quality was not significantly impacted by nitrogen fertility, but soil moisture stress had a significant effect on dough characteristics. Wet weather, which promoted grain deterioration, had the most significant influence on roti quality [38].

In addition to relationships between grain physical traits and roti quality, there were significant correlations between roti quality and the chemical properties of sorghum grain reported. Though it is not bread, Aboubacar and Hamaker [45] showed that sorghum couscous hardness correlated positively with the amylose content of flour. A positive correlation between the amylose content and overall acceptability of rotis made from sorghum flour was found [29,46]. Subramanian and Jambunathan [40] found a strong positive correlation between the texture of rotis and protein/total amylose of sorghum flour as well, along with a negative correlation between soluble sugars of sorghum flour and the texture of roti. In sorghum grain, it is known that more protein is found in the harder, more corneous outer layers of the grain (e.g., [47]). Fine flour fractions which comprised particles from the inner floury endosperm of sorghum grain have less total protein [40], thus resulting in an unwanted textural quality of roti [40]. This reflects the close relationship between physical properties, chemical properties, and milling of sorghum grains and how such relationships can influence the final quality of sorghum-based food products [40]. Chavan et al. [32] conducted a study on the nutritional quality of grain sorghum (post-rainy season) genotypes, which were developed through a systematic breeding program and compared with traditional genotypes. Many nutritional components including crude protein, total soluble sugars, soluble proteins, and soluble amylose content were found to be primarily important for high-quality roti (Table 1). Nandini and Salimath [33] reported that arabinoxylans were important in the quality of roti as they were responsible for the entrapment of gas in the dough; as a result, the final product had a softer texture. Although arabinoxylans from various cereals have the same basic chemical structure, the differences in the molecular features of arabinoxylans, which include the degree of branching, the spatial arrangement of arabinosyl substituents along the xylan backbone, and the ferulic acid content, can affect the viscoelastic properties of gels, resulting in changes in their conformation and causing arabinoxylan molecules to interact with each other and with other polysaccharides. The crispier texture of rotis, which are made from sorghum, compared to rotis prepared with wheat flour may be related to the highly branched nature of sorghum arabinoxylans, which form an inflexible matrix [33].

Nandini and Salimath [31] studied the carbohydrate profiles of wheat, sorghum, and bajra varieties with good chapati/roti-making quality and their isolated fractions such as water-soluble polysaccharides, barium hydroxide extract, hemicellulose A, hemicellulose B, and alkali-insoluble residue. In addition, the content of total sugar, uronic acid, rhamnose, fucose, arabinose, xylose, mannose, galactose, glucose of wheat, sorghum, and bajra varieties and their fractions were also measured. However, the researchers did not report on the effect of these components on the quality of the chapati/roti. To obtain optimal rolling quality, Chandrashekar and Desikachar [48] recommend choosing grains with a low gelatinization temperature, high peak viscosity, setback, and high water absorption [48]. Sorghum dough made with pre-gelatinized starch or flour from puffed grains had higher water absorption, and as a result, the dough handling and sensory properties of roti were improved compared to the control.

In terms of the flavor characteristics of roti, a positive correlation was found between flavor and protein, amylose, or ash content [40]. Carbohydrate composition and isolated fractions of sorghum, which are used for the good quality of roti, were investigated to determine relationships between taste and reducing sugars, water-soluble flour fraction, or flour swelling capacity, as well as the positive correlation between taste and water-soluble protein [40]. Unfortunately, mechanisms behind the correlated parameters, as well as why the chosen parameters were important for roti quality, were not explained. More studies need to be conducted to show the importance and mechanism of this research.

Due to the properties of sorghum flour and its impact on starch digestibility [49], there have been several research studies conducted on the glycemic index values or the digestive effects of sorghum-based foods (Table 1). However, there have been only a few studies on the glycemic index of sorghum roti. Prasad et al. [43] studied the glycemic index and glycemic load of different sorghum foods and compared them with those of wheat/rice-based foods. Interestingly, Prasad et al. [43] found that all sorghum-based products tested, with the exception of sorghum roti, had a lower glycemic index than their respective wheat/rice-based foods. The authors related the high glycemic index of the sorghum roti to the disruption of the outer layer of starch granules due to the processing of the roti. Nambiar and Patwardhan [50] studied the glycemic index and glycemic load data of the traditional millet-based recipes of India and found that cooking techniques such as shallow frying, roasting, and steaming significantly impacted the glycemic index. Similar to the findings of Prasad et al. [43], roti had a higher glycemic load compared to other food products tested. Roti has a short shelf life and will become dry within 10–15 h of cooking [36]. The shelf life of sorghum roti was studied in order to improve the storage of roti [25] (Table 1). Both ascorbic acid and potassium sorbate solutions were used in the formulation of roti to improve shelf life. Sorghum roti stored in polyethylene bags that contained either 0.5 g of potassium sorbate or 100 ppm of ascorbic acid in the roti formulation could be stored at room temperature for 6 days without any significant changes in sensory properties.

The impact of sorghum grain components on roti quality has been the subject of a wide range of studies, but the impact of processing methods on roti quality has received less attention. More research must be conducted on the use of various approaches to improve the quality of roti because there has only been a small amount of research on various processes and techniques.

### 3.2. Tortillas

Tortillas with alkali-cooked corn are commonly consumed in Mexico and Central America. However, sorghum or sorghum and corn blends are often used to bake tortillas because sorghum has a higher grain yield in hot and dry climates and is often a less expensive ingredient in certain Latin American countries [51,52]. Moreover, sorghum has a comparable nutritional value to corn [22,51,52]. Accordingly, substantial research has been conducted on using sorghum to produce tortillas using nixtamalized sorghum as well as the use of sorghum in place of wheat in flour tortillas (Table 2).

Nixtamalization is the traditional process used to produce corn flour used in tortilla making. In this process, corn is boiled in water with a low (1–5%) level of calcium hydroxide, soaked overnight, washed, and then ground in a stone grinder. The obtained flour is called “corn masa”. The masa can be wet or dried to obtain dry masa flour, which is also called an “instant” tortilla mix [52]. Masa is formed into dough balls, and the dough balls are flattened into thin layers and baked on both sides on a hot plate [68]. The alkaline treatment used during nixtamalization damages cell walls, allowing the pericarp to be removed more easily. The alkaline conditions also solubilize cell walls in the peripheral endosperm, induce swelling, partially destroy the starch granules, and change the physical appearance of protein bodies. In tortilla baking, cell walls are further degraded, starch crystallinity is lost, and protein bodies are partially destroyed [52] (Table 2).

Nixtamalization also reduces the total phenolic content of sorghum grains. Luzardo-Ocampo et al. [69] compared the influence of cooking or nixtamalization (alkaline cooking) on the bioaccessibility and antioxidant capacity of phenolic compounds of two sorghum varieties grown in Mexico (white/red). Variety, thermal treatment, and digestion phases all had a role in the changing bioaccessibility of phenolic compounds. Total phenolics and flavonoids became more bioaccessible after nixtamalization as well as after cooking. However, only nixtamalization resulted in a substantial decrease in the condensed tannins, and reduced tannin content by ~74%. The washing phase in the nixtamalization process resulted in the loss of pericarp and therefore phenolic compounds, which was stated as the reason for the lower phenolic content values in the processed samples. Importantly, nixtamalization impacted the bioavailability of the phenolic compounds present. For example, flavonoids from white sorghum showed strong absorption in the small intestine in the nixtamalized sample [69]. To find the best parameters for nixtamalization, different lime concentrations (0, 1, and 2%) and cooking times (20, 30, and 40 min) were tested by Gaytan-Martínez et al. [70] for white and red sorghum. Significant negative correlations between total phenol, flavonoid, and antioxidant capacities and nixtamalization parameters were found. The optimal conditions for maintaining antioxidant potential with low tannin content were 1.13% lime and 31.11 min of cooking time [70].

The majority of research on sorghum nixtamalization has focused on antinutrients and phenolic content. However, nixtamalization also improves sorghum protein bioaccessibility in tannin-containing sorghum by depolymerizing condensed tannins and disrupting protein–tannin complexes [71]. Interestingly, the bioaccessibility of phenolic compounds differed considerably throughout the digestive process, indicating that the release of condensed tannins from the food may also alter protein bioaccessibility and digestibility [69]. Cabrera-Ramírez et al. [71] conducted a study to evaluate how the nixtamalization method affects protein bioaccessibility in white and red sorghum varieties during in vitro gastrointestinal digestion. The results of this study showed that the nixtamalization process had no impact on the oral and gastric bioaccessibility of sorghum protein from both white and red sorghum, but it improved protein bioaccessibility during the intestinal digestion step, thus making nixtamalization a useful processing method for improving the nutritional quality of sorghum tortillas [71]. In addition to affecting digestibility, nixtamalization can impact protein composition. For example, Ali et al. [72] discovered that soaking sorghum in NaOH solutions at room temperature (27 °C) reduces the concentration of albumins and globulins.

Different levels of sorghum bran (0, 5, or 10%) have been added to corn tortillas before or after the extrusion process to improve the nutritional quality of extruded nixtamalized corn flour tortillas [60,61]. Total phenolic compounds and antioxidant activity levels in tortillas were lowered by the thermal technique used during baking. Corn flour with 10% sorghum bran added before extrusion retained almost 82 and 90% of the total phenolic compounds and antioxidant activity, while tortillas prepared with corn flour with 10% sorghum bran added after extrusion preserved more than 92 and 76% of the total phenolic compounds and antioxidant activity. The best texture, as well as the highest amount of total phenolic compounds and antioxidant activity, were obtained from tortillas prepared with sorghum bran added before extrusion. The retention of total phenolic compounds and antioxidant activity from flour to tortillas was higher for tortillas made with corn flour with 10% sorghum bran added after extrusion than for tortillas made with extruded nixtamalized corn flour [60].

In another study, researchers found that sorghum bran addition (5 or 10%) prior to extrusion increased free phenolic acid content and in vitro antioxidant activity [73]. Tortillas with sorghum bran added had higher total starch, amylose, and total soluble and insoluble dietary fiber content compared to the control (nixtamalized corn flour) due to the higher water absorption capacity of sorghum bran and its interaction with starch molecules from nixtamalized corn flour. Tortillas with sorghum bran had better texture and flexibility compared to those prepared without sorghum bran [64,65]. The addition of sorghum bran in corn flour tortillas also caused an increase in ferulic acid, cellular antioxidant activity, flavones, and total phenolic compounds [59,73]. Sorghum bran is a byproduct of sorghum dry milling and, in addition to containing fiber and phenolic compounds, also contains protein [74]. Thus, the addition of sorghum bran to corn tortillas could improve the protein content.

Tortillas prepared with whole white sorghum had a softer texture and darker color than those made with white corn [75]. Bedolla et al. [75] suggested that sorghum could be used alone or in combination with corn to make tortillas that were satisfactory in color, flavor, and texture. Quintero-Fuentes [55] reported that the use of sorghum provided more rollable and flexible tortillas compared to those prepared without heterowaxy sorghum. This might be more of a water-binding effect of waxy samples. In addition, grain from sorghum cultivars with a tan plant, white pericarp without sub-coat, intermediate endosperm texture, and low levels of color precursors is used to make tortillas with acceptable color and texture [55]. Research has demonstrated the benefits of using micronizing, which is a dry heat technique. Substituting micronized sorghum flour for commercial corn flour improved the palatability of the masa and the rollability of the tortilla. If micronizing is used to make tortilla flour, the process will be speedy and cost-effective because it will eliminate the existing method’s extensive heating and soaking times as well as the costly drying procedures [62,64] (Table 2).

The kernel size, texture, and structure of sorghum grain affected the quality of tortillas as much as they impacted the quality of roti. Khan et al. [56] reported that the lightest-colored tortillas with the highest potential quality were produced from sorghum with white or colorless pericarps and no testa, while colored sorghum produced tortillas with an undesirable color and poor taste. Sorghum flour from decorticated kernels was tested at varying levels of substitution for wheat flour [66]. There were three varieties of decorticated soghum: medium and fine. Sorghum flour was used to replace wheat flour at a rate of 15% to 30%. Mixing standards were followed for preparing composite doughs. When sorghum flour replaced 30% of wheat flour in wheat sorghum dough, the maximum stress peak, stress during relaxation, stress at a given strain, and dough viscosity all increased [66].

Sorghum hybrids and commercial sorghum flour were also evaluated to make gluten-free sorghum tortillas [57]. Tortillas with commercial sorghum flour had the highest sensory scores due to a smaller particle size and higher starch damage, which resulted in higher water absorption. Sorghum flour with smaller particle size and higher starch damage produced softer and more extensible tortillas [57]. The findings revealed that sorghum hybrids varied in kernel and flour quality, which could aid in the prediction of sorghum flour quality for gluten-free products.

### 3.3. Injera

Injera is an important food staple in Ethiopia, Eretria, and portions of Somalia [76,77]. Injera is a thin round bread with the top surface containing “eyes” and an overall soft and fluffy texture that can be rolled without breaking [56,57]. Injera is typically made from teff, and teff-based injera is said to have the best quality compared to other cereals [77,78]. In addition to teff, sorghum is also a common cereal grain for making injera [77,79]. The three main steps used in making injera are the preparation of a batter with flour and water, the addition of batter from the previous injera batter, and then fermentation with dough at an ambient temperature for almost 48 h. Then, a small amount of batter is poured onto hot clay and baked. A standard procedure for making injera was described by Yetneberk et al. [78,80], and the recent review of Neela and Fanta [77] details the history and traditional production of injera.

The production of injera from sorghum has been conducted by several research groups (Table 3), with reasons for this typically cited as desirable due to the cost of teff relative to sorghum and the low yield of teff [77,78,81]. Research efforts have been made to identify sorghum genotypes that have the potential to produce high-quality injera using both the instrumental characterization of injera and consumer preference sensory panels [81,82,83]. Yetneberk et al. [78] studied the influence of different cultivars on injera quality using different sorghum cultivars that varied in kernel characteristics. The findings showed that the quality of injera depended on sorghum cultivar and identified three sorghum cultivars that produced injera with positive sensory attributes and quality traits. The chemical (total starch, amylose, and tannin contents), image, and/or sensory analysis results of the study identified several sorghum genotypes that were superior to teff in making injera [84] (Table 3). The imaging analysis also allowed for a more objective and quick evaluation of injera [84]. Thus, the selection of sorghum lines with quality attributes specific to the production of injera may be one avenue to enhance the utilization of sorghum for injera production.

In addition to cultivar selection, other methods have been investigated to improve the quality of injera made with sorghum. Decortication and the use of sorghum-teff blends were also evaluated as techniques to enhance the quality of injera [80]. Decortication enhanced the color and other quality characteristics of injera by lowering the level of non-starch components in the grain. Both decortication and the use of a sorghum-teff blend were found useful to improve the quality of injera since the tannin content of sorghum was reduced. However, composing sorghum and teff rather than decorticating was found to be a better option since higher grain loss occurred during decortication [80]. Abraha and Abay [81] also reported that 50:50 blends of teff and sorghum produced acceptable injera. Ghebrehiwot et al. [76] also investigated blends of teff and a closely related grass species, *Eragrostis curvula* (Shrad.) Nees, with low levels of sorghum (5 and 10%), and reported that the texture, taste, and appearance of the teff injera were improved with the addition of 5% sorghum.

Thermal processing has an impact on the nutritional content of food products [85]. Accordingly, changes in the chemical and nutritional properties of sorghum flour, processed from flour into batter and into cooked injera, have been reported [86]. The content of protein, ash, and fat were reduced in injera compared to that of the flour and batter. The levels of antinutrients (polyphenols, phytate, and tannins) were reduced, and levels of Ca, Fe, Cu, and protein digestibility were higher in injera than in flour and batter [86].

**Table 3 foods-11-02466-t003:** The results of studies conducted on sorghum-based flat breads with preferment (injera and kisra).

Research Results	Reference
Substantial variations in injera texture, mouth feel, suppleness, and overall rate among the cereal flour blends (teff, barley, sorghum, and corn) were observed. No variations in injera in color, flavor, or the appearance of injera surface gas holes were observed.	[81]
Tannin levels were strongly associated with color and negatively associated with taste, whereas high starch content was associated with softness and rollability.	[84]
Eleven new sorghum varieties and eight promising experimental hybrids were compared for kisra-making quality with popular local varieties	[87]
Tannin had a negative effect on the protein quality and physical properties of kisra with tannin sorghums. White tan and non-tannin sorghum cultiivars were found more suitable for kisra production.	[88]
Tannins, phytic acid, and trypsin inhibitory activity in kisra bread were significantly decreased by traditional Sudanese kisra processing.	[89]
Kisra bread was improved with legume protein isolates in order to increase the protein content of Kisra and the amino acid profile, especially lysine.	[90]
A blend of teff (0.55%), sorghum (0.37%), and maize (0.07%) was found desirable and healthful, especially for people who lead sedentary lives and do not require much energy.	[91]
The addition of wheat bran to sorghum flour lowers sugar, in vitro protein, and starch digestibilities but increased crude fiber and the starch content of kisra.	[92]
By replacing 20% wheat and 10% pigeon pea with sorghum, changes in antinutrients and in vitro protein digestibility might well be obtained.	[93]
Brown bread for newborns (different types of kisra, some fortified with chickpeas or peanuts) should not include more than 10% wheat or sorghum bran, as this reduces digestion.	[94]
Waxy sorghum was recommended for making high-functional-quality gluten-free injera.	[95]
The addition of Monechma ciliatum seed flour to sorghum kisra significantly improved its nutritional value.	[96]
The overall acceptability of injera produced using disc mill flour was greater than that of hammer mill and blade mill flours. The injera produced using blade mill flour had the lowest level of rapidly available glucose and fast digested starch.	[97]
During kisra fermentation, the tannin level of the protein fractions was reduced, particularly in the albumin and glutelin fractions.	[98]
Fermentation increased the protein solubility, oil-binding capacity, emulsifying capacity, and emulsifying stability, while it decreased the water-binding capacity.	[99]
The addition of *Saccharomyces cerevisiae* to the prior starter (*Lactobacillus fermentum*, *Lactobacillus brevis*, and *Lactobacillus amylovorus*) reduced the sorghum fermentation time down to 4 h.	[100]
The microflora of Sudanese sorghum flour, a spontaneously fermented sourdough, and a long-term sourdough produced in a Sudanese household by consecutive reinoculations, was produced.	[101]
Traditional fermentation decreased soluble sugar content while increasing major by-products such as lactic acid, acetic acid, and ethanol.	[102]
Sorghum flour fermentation increased lysine and phenylalanine content while decreasing isoleucine, leucine, tryptophan, valine, methionine, and tyrosine levels. Baking reduced cystine and phenylalanine levels while increasing valine, isoleucine, and leucine content.	[103]
Fermented baobab fruit pulp flour as a starter decreased the antinutritional components of the fermented sorghum dough.	[104]
Kisra fermentation increased protein digestibility while having no effect on sample sensory quality. Sorghum can be supplemented with whey protein to increase its nutritional content and acceptability, even after baking.	[105]
Kisra fermentation resulted in a moderate improvement in protein, tyrosine, and methionine content, as well as a significant decrease in starch, total, and non-reducing sugars.	[106]
A nutritionally balanced Injera can be produced by fermenting 55% teff, 30% sorghum, and 15 % faba bean for 72 h.	[107]
Fermentation and heating increased the antioxidant potential of Tabat and Wad Ahmed sorghum grains.	[108]
During the fermentation of the Sudanese kisra dough, although many bacteria and molds grow, the bacteria with the highest population was P. pentosaceus.	[109]
The addition of 30% edible groundnut flour improved protein content and kisra quality significantly.	[110]
The main bacteria in fermented sorghum kisra and hulumur doughs were firmicutes and proteobacteria phyla.	[111]

### 3.4. Kisra

Kisra is a Sudanese bread made from sorghum, which is similar to injera but is smaller, thinner, and lacks a spongy texture [112]. Overall, the baking steps are similar to those of injera. The formulation for kisra can vary [87], but several procedures have been described in the literature (e.g., [87,88,100,102], Table 3). The processing of sorghum flour into kisra can significantly impact the final composition. For example, during the fermentation process of kisra production, the tannin, starch, total, and non-reducing sugars decreased while the protein, glutelin, crude fiber, tyrosine, and methionine content increased [98,102,106].

As with other sorghum-based foods, the sorghum cultivar used for kisra production has a significant effect on the final quality of the product. The evaluation of Sudanese sorghum cultivars during cooking and by sensory evaluation found that kisra quality varied among the sorghum samples [87]. Sorghum types with a floury endosperm were preferred over those with a corneous endosperm for cooking quality. For final production quality, kisra color and texture were found to be the most important traits, and kisra made from white sorghum types was rated the highest [87]. Awad-Elkareem and Taylor [88] also reported that non-tannin white sorghum types were the best for kisra production and could be a good option for the production of gluten-free wrap.

Research has been conducted to combine sorghum flour with other flours to improve the nutritional quality of kisra. For example, kisra made with sorghum flour plus bean protein isolates had higher protein and lysine content compared to kisra made only with sorghum [90]. The use of lactic acid bacteria starter cultures to reduce the fermentation time required for kisra production has also been investigated [100].

## 4. Pan Breads

### 4.1. Gluten-Free Pan Bread

Celiac disease, as the most prevalent genetic disorder in humans, is a food intolerance that lasts a lifetime. HLA DQ2 and DQ8 haplotypes are shown to contribute to the genetic factors of celiac disease. Although genes found in humans increase the likelihood of developing celiac disease, environmental factors have also been proven to have a role in the development of this autoimmune disease [113]. Individuals suffering from this condition should follow a gluten-free diet for the rest of their lives. In addition to patients with celiac disease, those having gluten sensitivity, or wheat intolerance, should follow a gluten-free diet [113].

Cereals (maize, oat, sorghum, rice, millet, teff, and oatmeal), pseudo-cereals (amaranth, quinoa, and buckwheat), legumes (peas, lentils, soybean, chickpea, and gram), seeds (flax seeds and pumpkin seeds), nuts (almond, walnut, and peanuts), tuberous rhizomes (tiger nut, jerusalem, and artichokes), and other types of raw materials (plantain and coconut) are used in gluten-free diets. As one of the gluten-free grains, the potential use of sorghum flour was evaluated in gluten-free pan bread [114,115]. As sorghum proteins lack the functionality of wheat gluten proteins and differ in other functional properties compared to wheat flour [114], substantial research has been conducted on methods to improve the functionality of sorghum for the production of pan breads. The germination of sorghum grain has been found beneficial in improving the functional properties of flour as well as enhancing the sensory and nutritional properties of cereals. The formation of exogenous and endogenous enzymes during germination decreased phytate content and improved bioavailability of iron and zinc in sorghum grains [115]. The activation of proteolytic enzymes, intrinsic amylases, proteases, phytases, and fiber-degrading enzymes caused a rise in nutrient digestibility [116,117,118]. Germination also increased protein solubility, foam ability, water and oil capacities, emulsion capacity, protease, and amylase activities in sorghum flour [117,118,119,120]. When germinated sorghum flour was used to make gluten-free bread, the firmness of the sorghum breads decreased, but cohesiveness increased with the increasing levels of germinated sorghum flour in the formulation [121].

In terms of the nutritional aspects, five different gluten-free breads (made from buckwheat, oat, quinoa, sorghum, or teff flour) were evaluated by Wolter et al. [122]. Among the breads tested, sorghum bread had the highest total available carbohydrates and the glycemic index of the sorghum-bread with sorghum was lower [122]. Additives used to improve the quality of sorghum bread have also been found to impact the nutritional composition of sorghum gluten-free bread. Combinations of potato starch–xanthan gum, rice starch–xanthan gum, and tapioca starch HPMC mixed with sorghum flour were evaluated for effects on sorghum bread quality, but also impacted fiber content, with the xanthan gum formulation having the highest total and insoluble dietary fiber content compared to the HPMC formulation [123].

The particle size of grain flour is highly important since it determines the available surface area for different types of reactions during bread making. Milling increases the exposure of starch to hydration and enzymatic action, and this results in increases in water absorption and starch digestibility of flours during dough making [124]. Schober et al. [124] also reported that starch damage in sorghum flour was related to sorghum bread quality and that differences in starch damage among the sorghum cultivars examined may have been due to differences in grain hardness. Thus, grain hardness variation and corresponding variability in starch damage of sorghum flour may be an important sorghum quality attribute for gluten-free pan bread production. Trappey et al. [125] investigated the effect of flour particle size on dough characteristics and final sorghum bread quality, particularly volume. Three different extraction rates (60, 80, and 100%) and pin milling at three different speeds (no pin-milling, low speed, and high speed) were used to mill sorghum grain into flour. Although a consistent relationship was not found between extraction level and starch damage, elevated starch damage was observed with the increasing speed of pin-milling. Bread with 60% extraction flour had the best specific volume, crumb characteristics, and toughness [125].

The blend of gluten-free flour, starches, and hydrocolloids are the core ingredients of gluten-free sorghum breads. Corn, tapioca, gelatinized tapioca, and potato starches are the most commonly used starches with sorghum flour [124,126,127,128,129]. Pre-gelatinized or native starches improve the quality of sorghum bread by increasing the consistency of the batter [130]. Research has demonstrated the positive impact of using the blend of decorticated sorghum flour and cassava starch or decorticated sorghum flour, gelatinized cassava starch, and raw cassava starch on bread quality [130]. The most common substitution level for starch is 30%. Since potato starch has a lower gelatinization temperature, which causes an earlier increase in the batter/crumb consistency during baking, compared to sorghum breads with 30% corn starch, breads with 30% potato starch were less likely to collapse [124]. The textural properties of gluten-free breads with five levels (10, 20, 30, 40, and 50%) of cassava, corn, potato, and rice starch supplemented sorghum flour samples were also evaluated for their impact on sorghum pan bread quality. Higher levels of starch substitution reduced the crumb firmness and chewiness while increasing the cohesiveness, springiness and resilience of all loaves. However, breads made with sorghum flour and 50% cassava starch had the best texture properties during storage [130]. The crumb properties of sorghum breads with native starch were found to be superior to those of pregelatinized starch [131].

Researchers have also studied the effects of the levels of fermented cassava, sweet potato, and sorghum flour on rheological and textural properties [132]. Breads with a minimum of 70% fermented cassava and up to 20% and 5% of sweet potato and sorghum, respectively, gave the optimum viscosity of batter and thus better bread quality, such as higher volume and better textural properties [132]. Whole grain flours (rice, sorghum, millet, amaranth, buckwheat, and quinoa) were also evaluated for their potential to produce gluten-free sorghum yeast rolls. Replacing the flour–starch base with whole grain flour produced acceptable gluten-free rolls. Furthermore, the fiber content increased by 2 to 5 times and the protein level doubled [133].

Hydrocolloids, along with starch and flour, are often used in gluten-free flour blends to enhance the rheological qualities of dough in the absence of gluten [134,135,136,137,138]. Hydrocolloids also enhance the quality of the fresh products and retard staling during storage. Thus, hydrocolloids, as gluten-substitutes, are the most commonly utilized ingredients in gluten-free bread formulations. Schober et al. [129] reported that xanthan gum deteriorated the crumb structure of gluten-free breads since it caused a decrease in gelatinization temperature, while HPMC (hydroxypropylmethylcellulose) improved crumb texture by increasing batter viscosity. While xanthan gum reduced loaf volume, skim milk powder reduced bread height by causing the loaves to collapse in the center [129]. Microcrystalline cellulose (MCC), carboxymethylcellulose sodium salt (CMC), methyl cellulose (MC), HPMC, and hydroxypropylcellulose (HPC) reduced resistance to dough deformation but did not cause any change in the crumb firmness and staling rate of the resulting bread [139]. In other research, the levels of corn starch/sorghum flour ratio, water, and HPMC blends were investigated [140]. In this research, the optimum formula for sorghum-based pan bread was found to be with a corn starch to sorghum flour ratio of 0.55 with 3% HPMC. One of the first detailed reports on the effects of hydrocolloids on sorghum-based pan bread was conducted by Hart et al. [126]. Arabic gum, carrageenan with vegetable mono and diglycerides, guar derivatives, HPC, MC and derivatives, sodium carboxymethyl cellulose, tragacanth, rapid-set pectin, slow-set pectin, low methoxyl pectin, and calcium were used to evaluate the effects of hydrocolloids in sorghum bread. Carrageenan combined with vegetable mono and diglycerides, guar derivatives, and tragacanth helped to prevent the bread from crumbling [126]. Each formulation of gluten-free bread had different levels and types of starch and hydrocolloid. A significant difference in the quality of loaves made with each starch/hydrocolloid combination was not found. Even though the role of batter viscosity in bread features was unexpected, in general, less viscous batters produced loaves with improved crumb grain characteristics. On the other hand, batter viscosity did not affect the loaf volume index [141]. Overall, hydrocolloids have been successfully used to enhance the quality of gluten-free sorghum bread, but their effects are mainly defined by the ingredients used in gluten-free formulation.

To evaluate the sensory properties (appearance, aroma, flavor, and texture/mouthfeel) of chemically leavened gluten-free sorghum bread with different starches and hydrocolloids, a lexicon with 28 attributes was developed by a panel highly trained in descriptive sensory analysis. The results showed bread with rice starch–xanthan gum or tapioca starch HPMC had a significantly better appearance, whereas bread with potato starch–xanthan gum and rice starch–xanthan gum had a significantly better texture/mouthfeel [123]. Gluten-free sorghum bread made from sorghum hybrids that varied in pericarp color was also evaluated by sensory analysis along with nutritional composition and antioxidant capacity [142]. Sorghum breads made from a tannin-containing sorghum had the highest antioxidant capacity but scored low in sensory analysis. Breads made from sorghum with a red pericarp (non-tannin) performed the best while still maintaining significant levels of antioxidant capacity [142]. Gluten-free sorghum bread made from tannin sorghum has been found to be classified as a low glycemic index, with a low overall carbohydrate content and increased fiber [143]; thus, additional research on improving sensory aspects of sorghum breads made from tannin sorghum is warranted.

Emulsifiers assist in the incorporation of air, the blending and emulsification of ingredients, dough stability, water absorption, texture and volume, and the extension of the shelf life of baked products [144,145]. Furthermore, in gluten-free bread formulations, the synergic interaction between hydrocolloids and emulsifiers has been extensively demonstrated [146,147,148,149,150,151,152]. Glycerol monostearate (GMS), sodium stearoyl-2-lactylate (SSL), calcium stearoyl-2-lactylate (CSL), and diacetyl tartaric acid esters of mono- and diglycerides (DATEM) all increased the elastic recovery of gluten-free sorghum dough and crumb softness while reducing the staling rate [139]. However, in another study, DATEM deteriorated the textural properties and specific volume of gluten-free sorghum breads [153]. The reason for the mixed results could be due to different ingredients in the two formulations between the studies. Glycerol monostearate, vegetable shortening, and mono- and diglycerides fatty acids caused a weakness in the crumb structures of bread with 100% sorghum flour [126].

Enzymes are widely used in baking and are highly effective in gluten-free systems [154,155,156]. Transglutaminase (TGase), an effective enzyme in the gluten-free industry, crosslinks proteins by catalyzing acyl transfer reactions that help to overcome the poor structure of gluten-free breads [157]. Different levels of microbial transglutaminase (0, 0.5, 1, and 1.5 U/g) were found to impact gluten-free sorghum batter properties and bread quality. Higher levels of TGase caused a decrease in resistance to the deformation and compliance of the batter, but an increase in zero-shear viscosity and elastic recovery. Furthermore, the crumb firmness and chewiness of the crumb increased as the level of the TGase increased, whereas a longer incubation time caused a decrease in crumb cohesiveness, chewiness, and resilience [158]. α-Amylases (1,4-α-d-glucan glucanohydrolases) are commonly utilized to improve the shelf life and the sensory and textural properties of wheat bread; accordingly, the function of α-amylases in gluten-free products has been investigated [131]. α-Amylases were found beneficial to sensory properties of yeasted gluten-free sorghum bread; however, their function in sourdough gluten-free sorghum bread was not fully understood due to the low pH of the batter [129]. Different levels of α-amylase (0–0.3 U g^−1^) were examined in gluten-free batter and bread with sorghum. The levels of α-amylase did not have any impact on the rheology of the batter, but higher levels of enzyme in sorghum bread caused a decrease in crumb firmness, cohesiveness, springiness, resilience, and chewiness, but an increase in adhesiveness [131]. In another study, bread with 100% sorghum flour, α-amylase, protease, or their combination (1 mg/200 mg flour) had weakened crumb structure [131]. Glucose oxidase creates arabinoxylans crosslinks as well as disulfide connections between proteins and a positive effect of glucose oxidase on the volume of gluten-free bread made from sorghum, as reported by Renzetti and Arendt [159]. Protein polymerization and higher protein phase continuity, as well as sorghum batter properties, were identified as factors for the improved bread quality. However, the negative impacts of protease treatment on the textural quality of sorghum loaves were demonstrated, with protein breakdown being the main factor [159]. The results of this study suggest that protein sources are important factors to consider when using enzymes that act on proteins to improve the bread-making performance of gluten-free flours.

Heat–moisture treatment, high-pressure treatment, extrusion cooking, etc., have also been used as innovative approaches to improve the quality of sorghum bread [160,161,162,163,164]. Heat application is used to improve the bread quality of weak flour by denaturing the proteins and inactivating the enzymes in flour, which improves the rheology and texture properties of dough and bread. Two temperatures (95 and 125 ºC) and three different durations (15, 30, and 45 min) were used to determine the heat effect on gluten-free sorghum bread [164]. Based on results, breads made from sorghum flour heat-treated at 125 ºC for 30 min had a higher specific volume, a more porous structure, better crumb characteristics, and a higher overall acceptability compared to control breads [164]. In addition, heat–moisture-treated cassava starch was also used as a functional in sorghum bread formulation. Compared to native starch, bread with heat–moisture-treated starch was softer. When the combination of heat–moisture-treated starch and amaranth malt were used in bread formulation, sorghum bread had higher cohesiveness values [162]. High-pressure treatment changes the composition of biopolymers, including proteins and starch, allowing for the production of foods, including breads, with new textures [160,165]. Vallons et al. [161] applied high pressure to sorghum flour, ranging from 200 to 600 MPa at 20 ºC and evaluated subsequent bread quality made from the treated flours. Batters made from flour treated at 200 MPa and 600 MPa were defined as weak and strong batters, respectively. In terms of rheological aspects, 300 MPa was a breaking point for batter consistency since the pressure between 300 and 600 MPa caused the gelatinization of sorghum starch, which improved the consistency of sorghum batter [160]. Thus, at pressures greater than 300 MPa, the pressure-induced gelatinization of starch made the batter stronger. Freeze-dried sorghum batters treated at 200 MPa and at 600 MPa replaced 2 and 10% of untreated sorghum flour in the bread formulation, respectively. The different levels of flour treated at 200 MPa did not cause any difference in bread quality, while different levels of sorghum treated at 600 MPa deteriorated the specific volume and increased the firmness [160].

Sourdough has been used as a processing technique for centuries, and it has gained popularity recently [166]. Sourdough is also an applied technique used to produce higher-quality gluten-free sorghum bread [134]. The use of sourdough technology caused the enhancement of the rheological properties of dough, resulting in improved texture, protein digestibility, antioxidant properties, and consumer acceptability [167,168,169,170,171]. Spontaneous fermentation was applied to sorghum bread with three different treatments, which were *Saccharomyces cerevisiae*, *Pediococcus pentosaceus*, and a mixture of both with control as breads with baker’s yeast. The control breads had a higher specific volume than sourdough bread. The height, texture, shape, crust color, and cracks did not change by sourdough application, but sorghum breads with baker’s yeast had lower scores in taste and overall acceptability [171]. The addition of 10% chickpea or cowpea flour to the blend of sorghum flour–corn starch (60:30%, w/w) decreased the firmness but increased the stiffness, gumminess, chewiness, fracture force, specific volume, ash, and L and b values of crusts of sorghum sourdough [168].

Another study was conducted to determine whether exopolysaccharides (EPSs) produced from sucrose during the sourdough process could replace hydrocolloids in gluten-free breads [172]. For this purpose, *Weissella cibaria MG1, Lactobacillus reuteri VIP,* and *L. reuteri Y2* were used for forming dextran, reuteran, and fructan, respectively, and also produced organic acids. EPSs were found to be more effective than organic acids in improving the quality of breads prepared with 10 or 20% sourdough. Breads and batters without sucrose and sourdough were used as controls in this research. Batters produced from sourdough had less strength and elasticity. Moreover, the degradation of sucrose to glucose and fructose increased the CO_2_ production of yeast during fermentation. Organic acids in the control of sourdough breads led to the hardening of the crumb. The impact of the organic acids was masked by the EPSs produced during sourdough fermentation, resulting in a softer crumb in both fresh and stored sorghum bread. Sourdough application did not significantly affect the specific volume of bread, except for breads with 20% sourdough, as demonstrated by *L. reuteri Y.* Overall, the results of this study showed that EPS formed during sourdough fermentation can be successfully applied to gluten-free sorghum flours to improve their bread-making potential. Schwab et al. [167] also compared the firmness of gluten-free breads produced with 14% sourdough addition. *Lactobacillus reuteri* LTH5448 or *Weissella cibaria* 10 M were used to ferment sorghum sourdough, producing fructooligosaccharides and levan, and isomaltooligosaccharides and dextran, respectively. The bread with *W. cibaria* 10 M was softer than bread prepared with L. *reuteri* LTH5448. Olojede et al. [169] produced sourdough with *Pediococcus pentosaceus* SA8, *Weissella confusa* SD8, *P. pentosaceus* LD7 and *Saccharomyces cerevisiae* YC1 to evaluate the structure of batter and bread and the nutrient properties of bread. Doughs with *Pediococcus pentosaceus* LD7 had the highest storage and loss moduli, whereas dough with *P. pentosaceus* SA8 sourdough had the lowest viscoelastic structure. Breads with *P. pentosaceus* SA8 and *S. cerevisiae* YC1 had the highest total dietary fiber but the least protein content. *P. pentosaceus* LD7 caused the most significant effect on the rheological properties [169]. The aggregation of sorghum proteins in the liquid phase of the batter during baking to produce strands and lumps, which interfered with the starch gel, may have caused the flat top and uneven crumb structure of sourdough sorghum breads (Figure 1).

Research has shown that the use of sourdough improves consumer acceptability, antioxidant properties, and the protein digestibility of gluten-free breads [170]. The addition of sourdough to dough made from 70% sorghum flour and 30% potato starch with 2% hydroxypropyl methyl cellulose improved the bread crumb by degrading peptides and making a stronger gel [129]. The addition of protein to bread may improve product quality factors such as flavor, texture, and storage stability.

Dairy and egg products have long been used in bread making because their proteins are highly versatile and easily incorporated into the dough. Milk proteins, as an example of animal protein, helped to improve crust browning but interfered with the starch gel by competing for water and disrupting the gel’s uniformity, resulting in a lower bread height, a higher bake loss, a lower crumb cohesiveness, and a collapsed top [124]. The addition of whole egg to sorghum formulation at 20, 25, and 30% levels created larger specific volumes, softer texture, better overall acceptability, crumb characteristics, and storage stability, but a darker crust [153]. As an example of plant-based protein, increasing the level of zein content improved the farinograph properties of sorghum dough and bread quality [173].

Ozone is a powerful oxidizing gas that occurs naturally. It can be used to disinfect food in direct contact and has been used in different applications, such as washing and tempering wheat, controlling insects and fungus in stored grain, as well as improving flour quality [174]. Ozone was applied at a frequency of 0.06 L/min for 15, 30, and 45 min. Crumb brightness and cell volume increased while the firmness decreased, but no change in specific volume was observed with an increase in the longer duration of ozone treatment [174].

### 4.2. Sorghum–Wheat Composite Pan Breads

Wheat has been a staple cereal in bread for centuries since wheat, protein, and gluten, are highly functional in the bread-making process. While bread made with sorghum flour is high in certain nutrients, sorghum alone does not contain the functional proteins needed for bread making [175,176,177]. Due to the inability of sorghum flour to produce a wheat-like viscoelastic dough, substantial research has been conducted using composite blends of wheat and sorghum flour. This research is especially important for regions of the world that do not produce large quantities of wheat and where wheat imports are expensive [176]. Sorghum flour, when incorporated at low levels with wheat flour (10% to 20%), has been found to create bread of comparable quality to wheat flour [177,178]. Four different levels (20, 40, 60, and 80%) of sorghum flour were used to evaluate the quality of wheat bread. The composite flour bread produced with up to 40% wheat flour was found to be acceptable [179]. At sorghum flour levels greater than 10%, the partial replacement of wheat flour with white sorghum flour changed the composition and rheological properties of the dough. The dilution of gluten takes place as the protein content is reduced, affecting the dough’s rheological behavior. A reduction in gas-retaining capacity was triggered by increased dough resistance and elasticity, which had an effect on bread volume. Despite the negative effects on physicochemical and rheological properties, bread made with the blend of wheat and white grain sorghum flour could result in satisfactory sensory quality [177]. Fine and coarse whole sorghum flours at levels ranging from 10% to 40% were replaced with wheat flour. The sorghum replacement improved the dough’s stability and increased development time. Sorghum additions at higher levels deteriorated the quality characteristics of bread, decreasing the volume and crumb porosity. There was much more depreciation when fine sorghum flour was utilized [180]. Another study evaluated the impact of three different levels of sorghum flour (5, 10, and 15%) and cysteine addition (30, 60, and 90 ppm) on wheat bread quality. The loaf volume and specific volume were reduced with the addition of sorghum flour and cysteine [181].

The majority of sorghum proteins are encapsulated within protein bodies, which renders the proteins unavailable for interaction during food processing and freeing the proteins from the protein bodies has been suggested as one method to improve the functionality of sorghum flour [182]. Sorghum germplasm has been identified with mutated protein bodies that may allow for increased interaction in the flour [183]. The composite wheat–sorghum dough made with a blend of high-digestibility high-lysine (HDHL) sorghum (sorghum containing more open protein bodies) and wheat had longer maximum resistance to extension and time to dough breakage at 35 °C compared to normal sorghum–wheat composite doughs. Bread with HDHL sorghum: wheat flour had lower firmness values, lower compressibility, higher specific loaf volumes, and higher springiness than normal sorghum–wheat composite bread [183]. The addition of 10, 20, and 30% of sorghum flour to wheat bread significantly increased the water absorption while decreasing dough stability. The sorghum replacement of more than 10% resulted in a cohesive dough with increased tenacity but decreased elasticity. There was a total loss of dough elasticity at 30% sorghum inclusion. There was a substantial decrease in bread volume when wheat flour was substituted with sorghum flour by more than 20%. However, no significant difference was found in the texture of composite breads [179].

The addition of gluten proteins has also been tested for its impact on sorghum–wheat composite flour and bread. Exogenous gluten protein considerably improved wheat–sorghum flour composite dough strength [175]. Water absorption, dough strength, and extensibility all decreased as sorghum flour content increased, while mixing time increased dramatically at set gluten protein levels. Exogenous gluten proteins could not form an effective gluten network for bread making when placed in the form of essential wheat gluten into the composite flour without wheat flour. The relationship between exogenous and endogenous proteins in wheat flour and the composite flour significantly influenced the rheological properties of the sorghum composite dough and hence bread volume [175]. The addition of zein to sorghum–wheat flour composite bread was also found to improve dough extensibility [184]. Xanthan gum and two different emulsifiers (SSL and DATEM) were also evaluated in sorghum and wheat composite bread. The combination of additives had a negative effect on bread volume. When xanthan gum was used alone, it increased the volume of bread. SSL and DATEM, on the other hand, softened the crumb and showed anti-stalling properties [185].

The extrusion cooking of sorghum has also been used in sorghum–wheat breads. Jafari et al. [186] investigated the use of sorghum flour extruded at 110 and 160 °C die temperatures with 10, 14, and 18% feed moisture on the physical and sensory properties of sorghum–wheat composite breads. The extrusion cooking of sorghum flour raised the bread crumb moisture, hardness, and redness; decreased the specific volume of breads; and gave it a denser crumb structure. However, the bread texture was enhanced by a reduction in the feed moisture and an increase in the die temperature. Extrusion cooking decreased the sandy properties of sorghum, enhanced the bread flavor, and thus enhanced the overall acceptability of extruded sorghum–wheat composite bread [186]. In another investigation, these authors demonstrated that extrusion cooking enhanced sorghum–wheat composited dough as well as bread quality through modifying the functional and structural properties of sorghum flour. Loaves with extruded sorghum flour had a lower number of cells/cm^2^, a lower total area, a lower pore area fraction (%), and a lower number of small cells (<2 mm^2^) compared to the control loaves. Nevertheless, the average pore size of the bread with extruded sorghum flour was smaller than that of the control bread. The addition of xanthan gum improved both the non-extruded and extruded sorghum wheat dough by increasing the water absorption and elasticity, but it had a negative impact on the textural properties of the final breads. Breads had harder textures when extruded sorghum flour and xanthan gum were used [187].

Sorghum grain was malted to decrease the gelatinization temperature and enhance the water-holding capacity of sorghum flour, hence overcoming the problems caused by the use of sorghum flour in composite breads [188]. Four different heat applications (drying the malt at high temperatures (50–150 °C), stewing, steaming, and boiling before drying the malt at 80 °C) were used to determine the effects of malted sorghum flour on the nutritional and technological properties of composite breads (70:30 wheat–sorghum). Amylases were inactivated by raising the malt drying temperature, but the malts became darker in color and bitter in taste. The amylases were entirely inactivated by steeping, steaming, and boiling the malt before drying, which enhanced the enzyme-susceptible starch concentration and paste viscosity of malt flours. In pan bread, boiling sorghum malt flour can be used as a partial substitution (30%) for wheat flour due to its enhanced crumb structure, water-holding ability, staling properties, and malt taste [188]. In another investigation, sorghum malting and wet-heat malt treatment were shown to lower the pasting temperature and increase the water retention capacity of sorghum [188]. The addition of different levels of dry nabag pulp powder (1, 3, 5, and 7%) and sorghum flour (10, 20, and 30%) fermented by *L. plantarum* and *L. brevis* in wheat–sorghum composite bread was tested. In terms of textural properties, bread with different levels of dry nabag (*Ziziphus lotus*) pulp powder and fermented sorghum flour had statistically similar textural properties, except bread with 100% wheat [189].

To evaluate the effects of high pressure on composite flours, high-pressure-treated (350 MPa, 10 min) sorghum flour was replaced with 40% of untreated wheat flour. Based on the results, high-pressure treatment provided better sensory scores and exhibited higher antiradical activities than those prepared by a conventional bread-making process and gluten incorporation. In addition, most physicochemical (staling kinetics and crumb grains) and nutritional features (relevant starch nutritional fractions and phenol contents) were not appreciably impaired by high-pressure treatment [190].

In terms of the nutritional properties, the addition of 20, 30, and 40% sorghum flour to composite flour and bread decreased gluten, protein, sugar, phytic acid, and essential amino acids (arginine, histidine, leucine + isoleucine, lysine, methionine, phenylalanine, threonine, tryptophan, and valine) but increased crude fat and ash [191]. Similarly, Sibanda et al. [177] revealed that the addition of 10, 20, and 30% of sorghum flour to wheat bread significantly decreased the protein and moisture content but increased the ash content.

In addition to functional and nutritional considerations, sensory analysis is an important part of consumer acceptance of any product, and therefore sorghum breads and sorghum composite breads. The rapid descriptive sensory descriptive techniques, check-all-that-apply (CATA) and optimized descriptive profile (ODP), were used by de Aguiar et al. [192] to distinguish gluten-free breads made with flours from various sorghum genotypes based on their sensory profiles, as well as to define the sensory properties of sorghum breads that impact customer liking. In this research, the appearance, color, texture, aroma, and flavor attributes of bread with two genotypes of white pericarp without tannin, bronze pericarp without tannin, or brown pericarp containing tannins (six different breads with six different genotypes in total) were studied [192]. Color and appearance-related features were found to be significant in CATA for distinguishing samples, while taste and texture descriptors were also significant in ODP. In both evaluation settings, panelists identified the best bread as being of bronze pericarp without tannin. Both techniques were effectively implemented and generated a common pattern of sample discrimination, but the attributes used for sample characterization and those defined as drivers of liking were dissimilar [193]. These findings could be used to guide the selection of sorghum cultivars for producing flour for sorghum–wheat composite breads. One of the most detailed and practical methods of sensory assessment is descriptive sensory analysis. A trained panel identifies and quantifies characteristics in the descriptive sensory analysis procedure. Several characteristics of sorghum and rye loaves were determined by descriptive analysis. The changes in formulation and processing were clear, but the findings revealed only minor differences between a commercially viable non-wheat bread and the composite bread. Slightly higher scores for sourness and astringency in the crust of sorghum composite bread than in rye bread were found. These findings suggested that up to 50% sorghum flour was acceptable in taste [192]. Levels of 10, 20, and 30% sorghum flour in wheat bread did not significantly affect the taste, flavor, and texture of the composite bread [180]. However, in another study, the sensory properties of composite bread decreased as the levels of sorghum flour increased from 20 to 30 and 40% [179]. Bread made with wheat flour (70%) and boiled sorghum malt flour (30%) had a smoother texture, was less prone to crumb firming, and had better flavor and chewing quality, as compared to bread made with untreated sorghum flour [188]. Furthermore, the addition of 10% of extruded sorghum flour improved the flavor and overall acceptability of bread [187].

## 5. Future Perspectives

It is critical to assess the rate of success for genetic improvement on a regular basis in order to find sorghum genotypes that may be employed in future breeding research to improve nutrient content and quality. Further research into the usage of preprocessing technologies should be carried out in order to improve the quality of sorghum flour and, as a result, the total product quality. Future research should also concentrate on the effects of the environment and crop management on sorghum quality. It is necessary to optimize the bread formulas and conduct a cost analysis. The comprehensive characterization of quality criteria should be the focus of future studies to improve high-sorghum breads (Figure 2).

## 6. Conclusions

Sorghum flour has been used to produce a wide range of bread and bread-like products, from “Western” style pan breads to “traditional” fermented flat breads and wrap-like products. As consumer preferences change, preferences for the style and type of bread products can also change. The sustainability and ability of sorghum to be grown in hot and dry climates, along with the unique health attributes found in sorghum flour, make sorghum an attractive flour for both gluten-free markets and functional food markets (including sorghum-only and wheat-sorghum composite products). Sorghum flour lacks functionality, and innovative processing techniques and technologies have been evaluated for improving the functional properties of sorghum flour and final product quality.

Sorghum flour and/or different types of flours can be used to make pan breads (gluten-free and sorghum–wheat composite pan breads) and flat breads (roti, tortilla, kisra, and injera). The most preferred grains in roti preparation are white/pale yellow, dense, and round grains. The pericarp color of sorghum grain, endosperm type, and texture significantly affect the quality of roti. The use of corneous grain up to a certain content improved dough and roti quality. There have been reports of the strong connections between roti quality and the chemical characteristics of sorghum grain as well as between roti quality and grain physical characteristics.

The quality of the tortillas was improved by using various percentages of sorghum flour. The quality of tortillas was influenced by the sorghum grain’s kernel size, texture, and structural composition. Both nixtamalization and cooking increased the bioaccessibility of the total phenolics and flavonoids.

Several sorghum genotypes were found to be superior to teff in making injera. Thus, the quality of injera was found to be dependent on sorghum cultivar.

As sorghum proteins lack the functionality of wheat gluten proteins and differs in other functional properties compared to wheat flour, substantial research has been conducted on methods to improve the functionality of sorghum for the production of pan breads. Sorghum grain germination has been proven to be advantageous for strengthening the sensory and nutritional qualities of cereals as well as the functional aspects of flour. The impacts of additives used to enhance the quality of sorghum bread on the nutritional value and quality of sorghum gluten-free bread have been well characterized. However, the substances used in gluten-free imitations largely determine their effects.

Variations in grain hardness and the resulting variations in starch damage in sorghum flour may be significant sorghum quality characteristics for the manufacturing of gluten-free pan bread. Replacing the flour–starch base with whole grain flour produced acceptable gluten-free rolls.

Sorghum flour has been proven to produce bread with quality similar to wheat flour when used in small amounts with wheat flour. Sorghum additions, however, at greater amounts, reduced the volume and porosity of the crumb, degrading the quality features of the bread. Although the physicochemical and rheological qualities were adversely affected, the sensory quality of the bread produced using the combination of wheat and white grain sorghum flour was acceptable.

Traditional processing methods contain many benefits for impacting sorghum flour and food quality, and these techniques should not be lost with changing consumer preferences. Combinations of novel processing techniques such as extrusion, high-pressure treatment, heat treatment, and ozonation with techniques such as fermentation may provide new avenues for improving sorghum flour functionality.

To investigate and develop processing techniques that improve sorghum flour functionality and end-product quality, emphasis needs to be placed on identifying sorghum germplasm with improved properties for the production of specific foods, along with breeding efforts to develop and improve sorghum for food functionality and quality.

Sorghum is known for being a genetically diverse crop, and several studies have identified sorghum germplasm that is superior for making specific foods. Selecting sorghum for food functionality and quality and improving nutrition for specific foods can impact significant portions of the world that rely on sorghum as a basic food staple.

## Figures and Tables

**Figure 1 foods-11-02466-f001:**
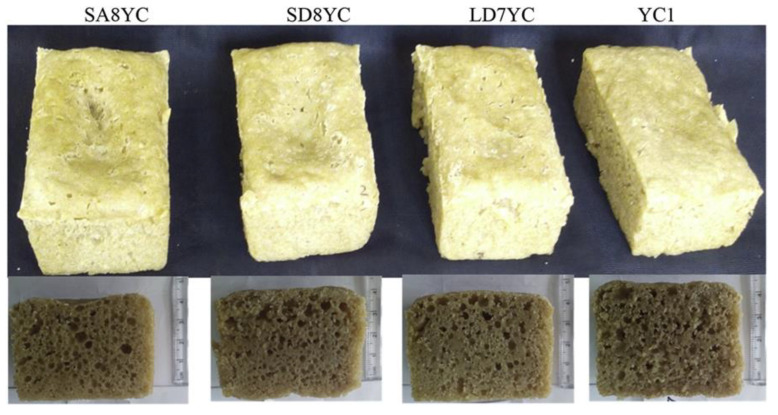
External appearance and internal structures of starter culture (*Pediococcus pentosaceus* SA8, *Weissella confusa* SD8, *P. pentosaceus* LD7, and *Saccharomyces cerevisiae* YC1) which produced sorghum sourdough breads. Reproduced from Olojede et al. [169]) with permission from the publisher.

**Figure 2 foods-11-02466-f002:**
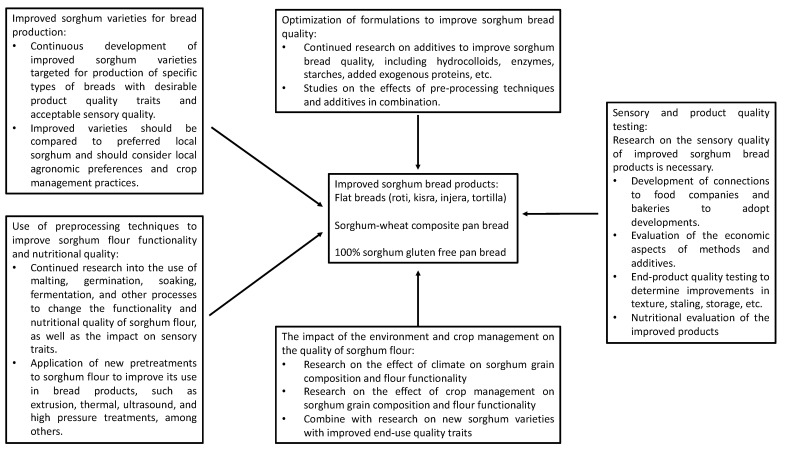
The future perspectives to improve high-quality sorghum breads.

**Table 1 foods-11-02466-t001:** The results of studies conducted on sorghum-based roti.

Research Results	Ingredients	Reference
Sorghum roti could be kept at room temperature for 6 days in polyethylene bags with either 0.5 g of potassium sorbate or 100 ppm of ascorbic acid in the roti formulation.	Sorghum flour, potassium sorbate, and ascorbic acid	[25]
Certain sorghum genotypes (CSH 16, SHD 36, CSH 15 and SHD 6, SPH 1864, and SPH 1867) developed through a systematic breeding program were reported to be promising for flour, dough, roti, and nutritional quality.	Sorghum flour	[26,27]
Significant differences in the overall proximate composition of nine improved sorghum types were found and one sorghum line (PVK 801) was identified as producing high-quality roti in terms of color, appearance, storage stability, aroma, and acceptability.	Teff, sorghum, and/or maize flour	[28]
In terms of nutritional quality and organoleptic parameters (color, appearance, flavor, texture, and overall acceptability), for roti quality, certain local land races and genotypes (SPV 1546, RSV 423) were found to be promising for protein, sugar, water absorption, and soluble protein content.	Sorghum flour	[29]
The tannin level in sorghum was found to have a modest influence on iron bioavailability.	Sorghum flour	[30]
The effect of these components on the quality of the chapati/roti was not reported. The hemicellulose B included similar levels of arabinose and xylose, while the hemicellulose A had more arabinose than xylose among the pentosans in the barium hydroxide extract of sorghum.	Sorghum flour	[31]
The high content of crude protein, total soluble sugars, soluble proteins, and soluble amylose was mostly responsible for the roti’s high quality.	Sorghum flour	[32]
Arabinoxylans were shown to be significant in roti quality because they were responsible for gas entrapment in the dough, resulting in a softer final product.	Sorghum flour	[33]
Germination and roasting of sorghum were found to impact the nutritional quality of the final roti.	Sorghum flour, millet flour, and/or amaranth flour, soybean flour	[34]
The traditional methods of milling, dough, and roti preparation were described in detail. No conclusion related to the evaluation of milling, dough, and roti preparation.	Sorghum flour	[35]
Sorghum flour with a specific particle size was required for desirable roti attributes such as extensibility and color.	Sorghum flour	[36]
To make good roti, the dough must be sufficiently cohesive and elastic.	Sorghum flour	[37]
Hard rotis with poor holding quality with grains containing 100% corneous endosperm were obtained. A weak dough, and rotis with poor flavor and preservation quality with floury grain varieties were obtained.	Sorghum flour	[38]
A strong positive correlation between the texture of rotis and protein/total amylose of sorghum flour was observed. A negative correlation between soluble sugars of sorghum flour and the texture of roti was found.	Sorghum flour	[39]
Higher levels of starch damage in sorghum flour resulted in a firmer roti texture and less acceptable among panelists in sensory evaluation.	Sorghum flour	[40]
The color, appearance, flavor, texture, taste, and overall acceptability were substantially altered as the proportion of sorghum flour replaced was increased.	Sorghum flour	[41]
Roti bread with sorghum-rich multigrain was negatively affected as the amount of sorghum flour increased.	Sorghum flour, wheat flour, ragi, black gram dhal, and fenugreek	[42]
All sorghum-based products tested, with the exception of sorghum roti, had a lower glycemic index than their respective wheat/rice-based foods.	Sorghum flour	[43]

**Table 2 foods-11-02466-t002:** The results of studies conducted on sorghum-based tortilla.

Research Results	Ingredients	Reference
The alkali weakened the cell walls and led to the swelling and partial loss of starch granules, altering the physical appearance of the protein structures.	Sorghum flour, calcium hydroxide	[52]
Two sorghum lines were identified that produced tortillas with similar quality as maize tortillas (both in terms of color and sensory properties).	Sorghum flour, calcium oxide solution	[53]
Tortillas made from the improved sorghum hybrids compared favorably to the positive control in the study.		[54]
Grain from sorghum cultivars with a tan plant, white pericarp without subcoat, intermediate endosperm texture, and low levels of color precursors gave tortillas with acceptable color and texture. The modification of cooking parameters, pearling levels, and corn-to-sorghum levels can result in more effective sorghum use, either alone or in combination with corn, for tortillas.	Sorghum flour, lime	[55]
Kernel size, texture, and structure all influenced sorghum cooking time and grinding characteristics. A good correlation was found between the alkali test and tortilla color.	Sorghum flour, calcium hydroxide	[56]
Sorghum hybrids’ kernel and flour properties differ, making it possible to assess the quality of sorghum flour for gluten-free food.	Sorghum flour, salt, xanthan gum, shortening, baking powder, citric acid, granulated sugar, monoglycerides, glycerin	[57]
Protein solubility and structure were found to be considerably altered when sorghum and maize were processed into tortillas.	Sorghum flour, calcium oxide	[58]
Sorghum bran supplementation increased bound and free hydroxycinnamic acids, flavones, total anthocyanins, and cellular antioxidant activity in corn tortillas.	Corn flour, sorghum bran, calcium oxide	[59]
Sorghum bran can be an effective option for increasing total dietary fiber content, lowering the predicted glycemic index, and improving the overall texture of maize tortillas.	Corn flour, sorghum bran, calcium oxide	[60]
Combining sorghum bran with extruded nixtamalized corn flour was critical for retaining antioxidants in tortillas and was sufficient for the production of novel functional cereal-based snacks with acceptable textural characteristics and color.	Nixtamalized corn flours, extruded nixtamalized corn, flours, sorghum bran	[61]
Extruded tortillas were less palatable than micronized or nixtamalized tortillas due to their brownish color and absence of an alkaline aroma.	micronized and extruded sorghum flour, calcium oxide	[62]
Thiamine, calcium, and amino acids (histidine, arginine, and leucine) were found to be the most influenced throughout the extrusion process.		[63]
Micronized sorghum absorbed more water, and the dough’s pliability improved. Tortillas with up to 20% micronized sorghum had acceptable color, taste, and texture.	Micronized pearled sorghum flour, alkali solution	[64]
As the amount of pearled sorghum in the blend increased, so did the cooking and steeping durations.	Pearled and whole sorghum flours, maize flour, calcium oxide	[65]
Sorghum flour (15 or 30%) in the wheat tortillas resulted in increased dough viscosity and increased stress in uniaxial tests.	Decorticated sorghum flour with different particle size, wheat flour, salt, citric acid, shortening, baking powder, potassium sorbate	[66]
The nutritional value of tortillas should not be affected by substituting sorghum for all or part of the maize.	Cooked or nixtamalized maize, whole sorghum, pearled sorghum	[67]

## Data Availability

Data reviewed in this article was all from published papers and book chapters, original data can be found in the cited publications.

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
