# Peer review of "Sorghum Flour Application in Bread: Technological Challenges and Opportunities"

_foods, 2022, doi:10.3390/foods11162466_

Round 1

Reviewer 1 Report

After reading the manuscript "Traditional and Novel Sorghum-Based Bread Products – A Literature Review",  I realized a great improvement in the quality of the paper.

The authors have accepted almost all of my requests.

They also have improved the English, which is always useful to ask a native speaker for a final appreciation. They improved and organized  the authors to better substantiate the methodology and corrected tables and Figures. This version of the paper is definitely much better.  I just have more 3 suggestions : improve the quality of figure 1 and 2;  The department in charge will guide you on this as well.

I missed the Methodology followed, how the study was conducted. Type of review? Keywords? Which databases?  Boolean operators? Inclusion and exclusion criteria at least.

Author Response

Dear Editor,

We appreciate the valuable comments of the referee on our manuscript of "Foods-1715741- Traditional and Novel Sorghum Based Bread Products-A Literature Review".  The suggestions were of great benefit for us to improve the quality of our paper. Based on the comments, we modified the manuscript, and the explanations for those changes are provided below. We hope the editor and reviewer will be pleased with our responses to their comments and the changes we've made to the original document.

Editor Comment 1:  After reading the manuscript "Traditional and Novel Sorghum-Based Bread Products – A Literature Review", I realized a great improvement in the quality of the paper. The authors have accepted almost all of my requests. They also have improved the English, which is always useful to ask a native speaker for a final appreciation. They improved and organized the authors to better substantiate the methodology and corrected Tables and Figures. This version of the paper is definitely much better. 

Response to Comment 1: We appreciate to the editor’s comment.

Editor Comment 2: I just have more 3 suggestions: improve the quality of figure 1 and 2.

Response to Comment 2: Thanks a lot for the editor’s comment. We improved Figure 2. We have tried to get high resolution image for Fig.1 from the journal.

Editor Comment 3: The department in charge will guide you on this as well.

Response to Comment 3: The department in charge got in touch with us. As soon as the manuscript is found to be suitable to be publish, we will pay charge.

Editor Comment 4: I missed the Methodology followed, how the study was conducted. Type of review? Keywords? Which databases?  Boolean operators? Inclusion and exclusion criteria at least.

Response to Comment 5: Thanks for the editor’s comment, we added a methodology part.

Reviewer 2 Report

Thank you for granting me the opportunity to review this piece of work. In this narrative review, Akin et al reviewed some traditional and novel sorghum-based bread products. Kindly, find below my comments for your response.

Title

The title is not clear and does not address the objective of the work. In the Abstract for example, at Line 15-17, the authors highlight processing methods that could be used to enhance the functionality of sorghum towards its utilisation in bread. Yet, this is not captured in the title and even in the text. The title should have been more of “Sorghum flour application in bread: Technological challenges and opportunities”.

Introduction

Line 25: Please, revise “total world production”

This review is really hard to follow. There is a general lack of direction in the write up. Consequently, the authors keep repeating information and much of the attention shifted towards the agronomic practices associated with sorghum cultivation than the bread. The authors fail to introduce the concept of bread and focus on gluten-free with sorghum as focus.

The authors also make several statements that are not supported by references.

General comments

The authors are trying to review the prospect of sorghum utilisation in different bread formulations. The direction of this review should have been a systematic review. The authors should have focused on “Sorghum flour application in bread: Technological challenges and opportunities”. That way, there is a clear focus that allows the authors to go deeper into the aim of the work. In Figure 1, the authors indicate that the title is “Table 1. The results of studies conducted on sorghum-based roti”. This warranted a systematic search. The authors should have created a column for other ingredients utilised. The dough preparation method used for example “straight dough” or otherwise. Processing methods that could be used to enhance the functionality of sorghum should for example have been a section title in this work. This review lacks depth and focus. Essentially, reviews need to highlight gaps in literature for potential research. However, I hardly see that in this review.

“Roti bread”

The authors should have had defined search terms that include: (sorghum) AND (Roti bread). That way, the outcome will be very specific. In this present form, the authors are even highlighting millet.

“Tortillas”

My comments for the roti bread apply to this as well.

Conclusion

The conclusion is too different from the objective of the work. I see the authors make general statements regarding other aspects of sorghum that are not really core highlights of this review.

Author Response

Dear Editor,

We appreciate the valuable comments of the referee on our manuscript of "Foods-1715741- Traditional and Novel Sorghum Based Bread Products-A Literature Review".  The suggestions were of great benefit for us to improve the quality of our paper. Based on the comments, we modified the manuscript, and the explanations for those changes are provided below. We hope the editor and reviewer will be pleased with our responses to their comments and the changes we've made to the original document.

Reviewer:

Reviewer Comment 1: The title is not clear and does not address the objective of the work. In the Abstract for example, in Line 15-17, the authors highlight processing methods that could be used to enhance the functionality of sorghum towards its utilization in bread. Yet, this is not captured in the title and even in the text. The title should have been more of “Sorghum flour application in bread: Technological challenges and opportunities”.

Response to Reviewer Comment 1: Thanks to reviewer for his/her valuable comment. We updated our title based on the reviewer’s comment.

Reviewer Comment 2: Line 25: Please, revise “total world production”.

Response to Reviewer Comment 2: We have updated the information based on your comment (page 1 line 25 “Sorghum (Sorghum bicolor L. Moench) is a grass belonging to the Andropogoneae tribe of the Poaceae family (alt. Gramineae) and is among the top five cereals in total world production after wheat, maize, rice and barley [1]).

Reviewer Comment 3: This review is really hard to follow. There is a general lack of direction in the write up. Consequently, the authors keep repeating information and much of the attention shifted towards the agronomic practices associated with sorghum cultivation than the bread. The authors fail to introduce the concept of bread and focus on gluten-free with sorghum as focus. The authors also make several statements that are not supported by references.

Response to Reviewer Comment 3: We have updated the introduction based on your comment. We revised the manuscript in a way to deliver the key findings highlighted in the manuscript.

Reviewer Comment 4: The authors are trying to review the prospect of sorghum utilization in different bread formulations. The direction of this review should have been systematic utilization authors should have focused on “Sorghum flour application in bread: Technological challenges and opportunities”. That way, there is a clear focus that allows the authors to go deeper into the aim of the work. In Figure 1, the authors indicate that the title is “Table 1. The results of studies conducted on sorghum-based roti”. This warranted a systematic search. The authors should have created a column for other ingredients utilized. The dough preparation method used for example “straight dough” or otherwise. Processing methods that could be used to enhance the functionality of sorghum should for example have been a section title in this work. This review lacks depth and focus.

Response to Reviewer Comment 4: We would like to remind that all the bakery products involve different methodology and the results only make sense with the details of the methods and findings. In this manuscript, we tried to give information about traditional and novel sorghum-based breads with the problems in developing such products as well as the advantages of the use of sorghum grain in those products. In the literature, many review studies show the use of sorghum flour in different kind of bakery products either on traditional or novel sorghum-based bread. This manuscript, however, appears to be a summary of all these observations. We believe this review covers a wide range of relevant published studies as well as fills a gap in the literature with its novel approach. It also reflects the differences in these results. As a result, we attempted to include the majority of relevant studies in the literature. We consider this creates the depth for the study.

In case of Tables, we added one column for the ingredients utilized. In terms of Roti bread, there was only one method to make roti dough also stated in manuscript. Thus, we did not include such a methodology.

Reviewer Comment 5: Essentially, reviews need to highlight gaps in literature for potential research. However, I hardly see that in this review. “Roti bread”

Response to Reviewer Comment 5: We have added research gaps in “Roti bread” section based on your comment. Please see page 7 line 200-203 (A wide range of studies has been conducted on the effect of components of sorghum grain on roti quality but limited research has been done to process techniques on roti quality there is only limited research on different processes techniques; therefore, more research needs to be done in using different techniques in roti quality).

Reviewer Comment 6: The authors should have had defined search terms that include: (sorghum) AND (Roti bread). That way, the outcome will be very specific. In this present form, the authors are even highlighting millet.

“Tortillas”

My comments for the roti bread apply to this as well.

Response to Reviewer Comment 6: We have changed the Table-2 based on your comment. We have added one column for the ingredients utilized in Tortilla Bread.

Reviewer Comment 7: The conclusion is too different from the objective of the work. I see the authors make general statements regarding other aspects of sorghum that are not really core highlights of this review.

Response to Reviewer Comment 7: Thanks to reviewer’s comment. We modified conclusion part.

Round 2

Reviewer 2 Report

Reviewer comments

Thank you for revising the manuscript. There are some revisions the authors can undertake to improve the quality of the review. Kindly, find mellow my comments for your response.

Line 78-79: the diabetes and obesity should be kindly revised to “anti-diabetic” and “anti-obesity”

Line 88-91: Please, provide a reference to support this

Line 102: Please, evise this “The term of…”. Kindly remove “of”

Line 124: kindly remove “etc” and complete the list of databases used

In Table 1, reference [166], the authors failed to mention the organoleptic attributes of the bread

In Table 1, the authors should summarise the findings of the work

Ref. [34], the authors stated just objective of the work “Carbohydrate composition of cereal varieties (wheat, sorghum, and bajra) having good roti making quality was evaluated”, which is not important

At ref. [24], [33], [37] and [175] why is the ingredient undefined?

Line 235: Pericarp colour of what?

General comments: I kindly suggest that the authors present a “PRISMA” guideline to illustrate the article selection process described in the “Methodology” section

In Table 1. I expect that the authors indicate the substitution ratio and indicate the organoleptic attributes. I think the authors should divide the Table and group the findings into organoleptic attributes, and also the properties of the flour

I kindly suggest that, the authors divide the writing under each bread type into two parts- the first part handles the effect of sorghum incorporation into the bread on the dough properties and the second parts investigates the effect of the sorghum use in the breads on the sensory properties.

I also suggest that, the authors create a new section title on “The effect of processing method on the functional properties of sorghum flour”. This is important because the authors have captured that in the Abstract.

The authors should kindly revise the Abstract. The aim of the review should be stated right after the “Introduction” of the Abstract.

Author Response

Reviewer comments

Reviewer Comment 1: Line 78-79: the diabetes and obesity should be kindly revised to “anti-diabetic” and “anti-obesity”

Response to Comment 1: We would like to thank reviewer. We have changed the lines based on his/her comment.

Reviewer Comment 2: Line 88-91: Please, provide a reference to support this.

Response to Comment 2: We added the references.

Reviewer Comment 3: Line 102: Please, revise this “The term of…”. Kindly remove “of”

Response to Comment 3: We would like to thank reviewer for noticing this mistake. We have changed the words based on your comment.

Reviewer Comment 4: Line 124: kindly remove “etc” and complete the list of databases used

Response to Comment 4: We removed “etc.“.

Reviewer Comment 5: In Table 1, reference [166], the authors failed to mention the organoleptic attributes of the bread.

Response to Comment 5: The organoleptic attributes was added.

Reviewer Comment 6: In Table 1, the authors should summarise the findings of the work

Response to Comment 6: We would like to thank reviewer. We modified Table considering present reviewer and previous reviewers’ comment.

Reviewer Comment 7: Ref. [34], the authors stated just objective of the work “Carbohydrate composition of cereal varieties (wheat, sorghum, and bajra) having good roti making quality was evaluated”, which is not important

Response to Comment 7: We deleted this sentence.

Reviewer Comment 8: At ref. [24], [33], [37] and [175] why is the ingredient undefined?

Response to Comment 8: We would like to thank the reviewer for noticing this mistake. We added the missing information.

Reviewer Comment 9: Line 235: Pericarp color of what?

Response to Comment 9: The information has been added.

Reviewer Comment 10: I kindly suggest that the authors present a “PRISMA” guideline to illustrate the article selection process described in the “Methodology” section

Response to Comment 10: We would like to thank the reviewer for his/her valuable suggestions. The current manuscript is a general review of the literature and was not a formal metanalysis. We made an effort to explain the procedures we followed when choosing the papers. However, it can be misunderstood as a systematic review and lead to confusion and may mislead readers. As a result, we removed several sentences from the methodology section that might be misunderstood. We have also looked at several recent reviews published in Foods to see how methodology has been handled in other reviews. Specific methods have not been published in any of these, so we feel the info included in the current version matches (or goes beyond) methodology standards of other reviews in Foods.

https://www.mdpi.com/2304-8158/11/14/2065/htm

https://www.mdpi.com/2304-8158/11/14/2064/htm

https://www.mdpi.com/2304-8158/11/14/2051/htm

https://www.mdpi.com/2304-8158/11/13/1977/htm

https://www.mdpi.com/2304-8158/11/13/1953/htm

https://www.mdpi.com/2304-8158/11/13/1924/htm

https://www.mdpi.com/2304-8158/11/13/1901/htm

https://www.mdpi.com/2304-8158/11/13/1867/htm

https://www.mdpi.com/2304-8158/11/13/1854/htm

Reviewer Comment 11: In Table 1. I expect that the authors indicate the substitution ratio and indicate the organoleptic attributes. I think the authors should divide the Table and group the findings into organoleptic attributes, and also the properties of the flour. I kindly suggest that, the authors divide the writing under each bread type into two parts- the first part handles the effect of sorghum incorporation into the bread on the dough properties and the second parts investigates the effect of the sorghum use in the breads on the sensory properties.

Response to Comment 11: We would like to thank reviewer for his/her suggestion. It should be noted, nevertheless, that not all studies underwent sensory analysis, and in only a small number of them, was sensory analysis employed. As a result, we made an effort to include all the studied technological aspects of roti in this table.

Reviewer Comment 12: I also suggest that, the authors create a new section title on “The effect of processing method on the functional properties of sorghum flour”. This is important because the authors have captured that in the Abstract.

Response to Comment 12: We would like to thank the reviewer for his/her suggestion. We modified the manuscript based on two sections: ‘Flat breads’ and ‘Pan breads’ which have different subsections. Each of these subsections also includes the effects of processing methods on the functional properties of sorghum flour. Information on preprocessing has been included where appropriate throughout the manuscript, but for each type of bread research in this area varies tremendously and so subsections for each type of bread have not been included.

Reviewer Comment 14: The authors should kindly revise the Abstract. The aim of the review should be stated right after the “Introduction” of the Abstract

Response to Comment 13: The abstract has been modified to include the overall aim of the review article.

This manuscript is a resubmission of an earlier submission. The following is a list of the peer review reports and author responses from that submission.

Round 1

Reviewer 1 Report

Sorghum is a cereal deserving interest now and for the future and it deserves research to improve its main issues.

The Review is well written and interesting and it is proposed with a different focus than those previously published on sorghum.

However, I suggest to strongly revise the organization of the manuscript. The tables reporting all the works considered in the Review on different sorghum-based products are poorly readable. Please find another way to present the articles in the Tables (maybe grouping by the main research activities). In the same way, the text is abundant, too long. Please try to divide the different paragraphs coherently to the Tables.

A summarizing conclusive Figure is welcomed. 

Author Response

We would like to thank the reviewer. We have updated the manuscript and tables based on the reviewer’s comment.

Reviewer 2 Report

After reading the manuscript "Traditional and Novel Sorghum-Based Bread Products – A Literature Review", I realized that the manuscript showed in some parts the scientific rigour wanted, but in other parts I have missed it.The authors have presented critical evaluation only in some paragraphs.The references are not exactly current, besides the objective could be more attractive and cientific.Thats why I have written some suggestions in an attempt to improve them.

L.25- I miss an author for this statement.

L. 44 This sentence: "Sorghum is similar to other cereal grains in overall proximate composition." Choose a better moment to include it.

L.49- Check it as well.

L.53 and L.58 - "sorghum grain is also known to contain bioactive compounds "- They are too similar. Improve this paragraph, please. 

L.66- 68- I think it is important to point out that raw sorghum is not usually consumed.

L.73-74- It should come close to line L.66-68 to continue the thought

L.80 - The agronomic part has already been mentioned above. Please follow the thought

L.82-83 - "sorghum proteins lack the unique functional
properties of wheat proteins in terms of viscoelastic dough formation." That information was not mentioned before, perhaps it could be insert something close to the protein line (L.72). You only mentioned that the proteins are poor, but you didn't comment on the technological performance.

L. 87- Before Flat breads the authors did not make clear the rationale and the objective of the paper.

L.89- I missed the Methodology followed, how the study was conducted. Type of review? Keywords? Which databases?  Boolean operators? Inclusion and exclusion criteria at least.

L.102- In my opinion, more important than the objectives of the studies are the results found by the authors.

L.160- I think that the information about the chemical part should come together in the Flat bread, the same about the technological parameters and sensory parameters. So that it doesn't become a back and forth.

L.349 and L. 363- these percentages that you mention in these lines are super important for other researchers to know the maximum levels already tested and up to which % better technological and sensory quality were observed, mainly.

Review papers are denser and heavier if they have a lot of text, as is the case of yours. Therefore, I would also suggest the addition of at least one figure or diagram. I think it would be more attractive to the readers. 

I suggest a table with these percentages, it would be very relevant for the readers of your article.

L.456- In this line you comment about the association with teff, it would be interesting to address other successful associations with sorghum in the literature, and obviously the maximum amount suggested by the researcher.

L.502- Why gluten-free bread at first? As these are special, I think it would be interesting to start with the most usual. 

L.509- It sounds to me like you missed mentioning some of the allowed cereals and suddenly included sorghum as one of them.

L.514-515-  "As sorghum proteins lack the functionality of wheat gluten
proteins and differs in other functional properties compared to wheat flour"- Since this paragraph is about Gluten-free pan bread, I don't think it would be the best moment to mention the quality of wheat, since it is a grain that will not be able to be utilized anyway.

L.529- Again, more important than the objectives of the papers, in my opinion would be the chemical, technological and sensory results of the elaborated breads, especially in the case of gluten-free products.

L.537-I suggest avoiding this type of information in parentheses "(discussed in more detail below)", and organizing the text in such a way that the paragraphs are close together.

L.583- Please, cite which researchers.

L.605- Onyango et al., 2009)-  Please adjust the citation to the journal's guidelines.

L.666-729- I wonder if these paragraphs should be in a separate subtitle, I don't see them specifically related to gluten-free products either. 

L.786- Now I found what I was looking for previously, I think this subtitle should come before the gluten free options.

L.789- Is the maximum percentage that does not compromise the three important axes: chemical, technological and sensorial quality?
What about the % of sorghum when it is used exclusively in pan bread? Is 100% sorghum viable?

L.952- The conclusion is about your research, you can not cite other authors in it.

Author Response

(The authors gave the same response as above.)

Round 2

Reviewer 2 Report

After another evaluation of the manuscript, I realized some improvement in the quality of the paper. The authors have accepted some of my requests.

English needs to be evaluated by a native speaker before the final appreciation. Methodology is still poor lacking details, tables and figures.